# BdCV1-Encoded P3 Silencing Suppressor Identification and Its Roles in *Botryosphaeria dothidea*, Causing Pear Ring Rot Disease

**DOI:** 10.3390/cells12192386

**Published:** 2023-09-29

**Authors:** Shanshan Li, Haodong Zhu, Ying He, Ni Hong, Guoping Wang, Liping Wang

**Affiliations:** 1College of Plant Science and Technology, Huazhong Agricultural University, Wuhan 430070, China; ss1144256452@163.com (S.L.); zhd2021301110160@webmail.hzau.edu.cn (H.Z.); yhe2022@webmail.hzau.edu.cn (Y.H.); whni@mail.hzau.edu.cn (N.H.); gpwang@mail.hzau.edu.cn (G.W.); 2Key Laboratory of Plant Pathology of Hubei Province, Huazhong Agricultural University, Wuhan 430070, China

**Keywords:** *Botryosphaeria dothidea*, RNA silencing, BdCV1, RNA silencing suppressors

## Abstract

Pear ring rot disease is an important branch disease, caused by *Botryosphaeria dothidea*. With the discovery of fungal viruses, the use of their attenuated properties for biological control provides a new strategy for the biological control of fungal disease. RNA silencing is a major antiviral defense mechanism in plants, insects, and fungi. Viruses encode and utilize RNA silencing suppressors to suppress host defenses. Previous studies revealed that Botryosphaeria dothidea chrysovirus 1 (BdCV1) exhibited weak pathogenicity and could activate host gene silencing by infecting *B. dothidea*. The aim of our study was to investigate whether BdCV1 can encode a silencing suppressor and what effect it has on the host. In this study, the capability of silencing inhibitory activity of four BdCV1-encoded proteins was analyzed, and the P3 protein was identified as a BdCV1 RNA silencing suppressor in the exotic host *Nicotiana benthamiana* line 16c. In addition, we demonstrated that P3 could inhibit local silencing, block systemic RNA silencing, and induce the necrosis reaction of tobacco leaves. Furthermore, overexpression of P3 could slow down the growth rate and reduce the pathogenicity of *B. dothidea*, and to some extent affect the expression level of RNA silencing components and virus-derived siRNAs (vsiRNAs). Combined with transcriptomic analysis, P3 had an effect on the gene expression and biological process of *B. dothidea*. The obtained results provide new theoretical information for further study of interaction between BdCV1 P3 as a potential silencing suppressor and *B. dothidea*.

## 1. Introduction

Pear ring rot disease, mainly caused by *Botryosphaeria dothidea*, causes canker and ring symptoms on the pear branches and fruits, and leads to tree debilitation and even orchard destruction [1,2,3]. It is particularly important to find a feasible green control measure, and the use of hypovirulent strain for control has gradually developed into an effective measure. Fungal viruses are widespread in all types of fungi and are transmitted during cell division, sporogenesis or mycelial fusion [4], even transmitted by insects [5]. It was first reported in *Agaricus bisporus*, which laid the foundation for the study of Mycovirology [6]. Subsequently, mycoviruses were found in *Penicillium chrysogenum*, *Saccharomyces cerevisiae* and *Aspergillus fumigatus* [7,8]. Some fungal viruses with weak pathogenicity characteristics have potential biocontrol ability and can be effectively used for biological control [9,10,11,12,13,14,15].

Our team previously identified *Botryosphaeria dothidea* chrysovirus 1 (BdCV1) as a weak pathogenic factor of *Botryosphaeria dothidea*. BdCV1 genome was determined to indicate that it was a new member of the Chrysovirus, containing four dsRNAs, dsRNA1 encodes RdRp protein, dsRNA2 encodes CP protein, and dsRNA3 and dsRNA4 encode putative proteins, respectively [16]. A library of *B. dothidea* infected with chrysovirus BdCV1 was constructed, the DEGs were mainly involved in metabolic pathways and secondary metabolic pathways of biosynthesis, and were also involved in cell replication, signal transduction and other disease-resistant pathways [17]. Several key components participating in gene silencing pathway are up-regulated expression under viral infection, suggesting that BdCV1 may trigger gene silencing in host [18].

Host fungi influence mycoviruses through RNA silencing, an antiviral mechanism that regulates gene expression after transcription [19]. Among RNA silencing pathways were three major conserved proteins: Argonaute (AGO), Dicer-like (DCL) and RdRp. DCL protein, as an endonuclease, cuts single-stranded RNA into microRNA (miRNA) or dsRNA into small interfering RNA (siRNA). A strand of siRNA forms a silencing complex (RISC) with the AGO protein and so on, and guides RISC targeting mRNA. RdRp protein is involved in amplifying silencing signals by synthesizing dsRNA [20,21,22]. Viruses usually encode different silencing suppressors in co-evolution with their hosts to block RNA silencing mechanisms in order to inhibit host gene silencing [23]. RNA silencing suppressors have been extensively studied in plant viruses, and they are involved in almost all steps of the RNA silencing pathway. For example, tombusvirus P19 [24,25], potyvirus HC-Pro [26] and beet yellows virus P21 [27] block the initiation step of silencing by isolating siRNA. Letevirus P0 can promote AGO1 degradation [28]. In addition to inhibiting RNA silencing, most plant virus silencing suppressors are also involved in other functions, such as pathogenicity, replication, intercellular movement and hypersensitive reaction [27,29,30,31,32]. Based on the research of plant viruses, some resources of fungal virus silencing suppressor have been explored. The most in-depth study is the multifunctional protein p29 encoded by CHV1, which inhibits host RNA silencing by inhibiting the transcriptional activation of *dcl2* and *ago2* [33]. p24 protein encoded by Cryphonectria hypovirus 4 (CHV4) acts as a silencing suppressor to promote stable infection of co-infected Rosellinia necatrix mycoreovirus 2 (MyRV2) [34]. Fusarium graminearum virus 1 (FgV1) -ORF2 with DNA binding ability inhibits the transcriptional up-regulation of key enzyme genes of RNA silencing, namely FgDICER2 and FgAGO1, to restrict the host’s antiviral defense mechanism [35]. The FgHV1 encoding protein p20 binds sRNA and has been identified to have the function of silencing suppressor in *Agrobacterium* infiltrate assay, which suppress not only local silencing but also systemic silencing [36]. VP10 of Rosellinia necatrix mycoreovirus 3 (MyRV3) is a silencing suppressor, and how it functions in fungal cells is unclear [37]. In conclusion, fungal silencing suppressors affect host gene silencing mainly by inhibiting the expression of RNA silencing component proteins. In addition, p29 and VP10 also have silencing suppressor activity on the heterologous host *Nicotiana benthamiana* [33,37].

At present, the main control methods for pear ring rot are bagging treatment and chemical control with a negative impact on the environment [38,39]. The discovery of the attenuated strains of *B. dothidea* provides a new strategy for disease control. It is very important to study the function of the protein encoded by mycoviruses. BdCV1 caused attenuated virulence of *B. dothidea*. In this study, BdCV1 silencing suppressor was screened and identified by agro-infiltrating assay, and its effect on host was analyzed to reveal the involvement of silencing suppressor in host defense response and gene silencing pathway. The results are intended to provide theoretical basis for virus-host interaction and provide new strategies and molecular information for fungal viruses to be used in biological control of pear ring rot disease.

## 2. Materials and Methods

### 2.1. Botryosphaeria dothidea Strains and Planting Materials

The *B. dothidea* strain LW–C infected with hypovirulent mycovirus-Botryosphaeria dothidea chrysovirus 1 (BdCV1), was derived from LW-1. The *B. dothidea* strain LW–VF derived from LW–C strain [40].

The seeds of *N. benthamiana* and *N. benthamiana* line 16c with an expressed GFP protein were retained and propagated by our laboratory. Plants were cultured in the growth chamber at 25 °C. The annual pear branches of “Yuanhuang” and “Jinshui No. 1” were collected from national fruit tree germplasm Wuchang sand pear nursery and fruit tree Practice teaching base of Huazhong Agricultural University.

### 2.2. BdCV1-Encoded Proteins Sequence Amplification and Expressed Vector Construction

The four ORFs (encoding RdRp, CP, P3 and P4) of BdCV1 genome were individually amplified from the strain LW-C. Then they were inserted into the pMD18-T (Takara, Dalian, China) vector for screening positive clone and sequencing analysis to identification, which were used for vector construction as the followings.

For silencing suppression activity analysis, the full-length ORF of RdRp and P3 were cloned into pCNF3 vector to produce pCNF3-RdRp and pCNF3-P3, respectively. The full-length ORFs of CP and P4 were connected to pCNF3 vector using double digests to obtain pCNF3-CP and pCNF3-P4, respectively. The obtained 4 constructed vectors were individually transformed into *Agrobacterium tumefaciens* GV3101, respectively.

In order to study the influence of P3 on host biological characters, the overexpression vector of P3 was constructed. The full-length ORF of P3 was linked into pDL2 vector to produce pDL2-P3 for fungal protoplast transformation. The primer information used for fragment amplification and vector construction is shown in Appendix A.

### 2.3. Agrobacterium-Mediated Transient Transformation Test

Taking 10 μL of *Agrobacterium* solution that has been identified as positive to 4 mL of kanamycin and rifampicin liquid LB 10 mL EP tube was incubated at 28 °C 200 r/min for 18–24 h. The bacteria are collected and rinsed. The final collected *Agrobacterium* was suspended in a solution (10 mmol/L MgCl_2_, 150 μmol/L AS, 10 mmol/L MES, pH 5.7) and incubated in a 28 °C incubator for 2–4 h in the dark. For silencing suppressor identification test, the equal volumes of *A. tumefaciens* containing P3 encoded protein and 35S-GFP, followed by infiltration into the leaves of *N. benthamiana* plants or *N. benthamiana* line 16c at 4–6 weeks. For analysis of dsRNA-induced silencing, the vectors were obtained 35S-GFP and 35S-dsGFP, respectively, as a mediator to induce RNA silencing. The fluorescence signal was observed with handheld UV lamp, and taken photographs to record the results.

### 2.4. The Total Proteins Were Extracted for Western Blot Analysis

Western and IP cell lysates (Beyotime, Shanghai, China) was used to obtain the total protein. The protein extracted was mixed 1:1 with 2× loading buffer solution, which was denatured. The loading of sample to be tested was 10 µL and the Multicolor Prestained Protein maker (Epizyme, Shanghai, China) was 5 µL in every gel lane for electrophoretic analysis. The gels in Coomassie brilliant blue (CBB) solution were stained, and further used for Western blotting analysis. The Rubisco large subunit with CBB staining was used as protein loading controls. The protein transferred to PVDF membrane was performed by wet transfer method. The anti-GFP rabbit polyclonal antibody and HRP-labeled Goat anti-Rabbit IgG (Sangon Biotech, Shanghai, China) were used for detection at a 1:5000 dilution, respectively. Western Bright ECL HRP (Advansta, San Jose, CA, USA) color solution was added to the surface of the PVDF membrane, which placed at room temperature and incubated away from light for 3 min. The ChemiDocTM Touch gel imaging system (Bio-Rad, Hercules, CA, USA) was used to take pictures.

### 2.5. Trypan Blue and DAB Staining

The leaves prepared for dyeing were cleaned and spread on a petri dish. After pouring Trypan blue or DAB staining solution, a filter paper covered the leaves to fully absorb the staining solution. The leaves were left for staining for 5 h in the dark, and then slightly cleaned with clean water. Then the leaves were poured into anhydrous ethanol and decolorized in boiling water bath for 10 min. The results were observed and recorded.

### 2.6. Protoplast Transformation

Virus-free *B. dothidea* strain LW–VF was cultured on the PDA at 25 °C in dark. The mycelium collected and cultured for 2 days was added to the lysis solution and shook for 3 h at 90 rpm. Hemocytometer was used to achieve a protoplast concentration of 10^7^ number/mL. The pDL2-P3 plasmid was transferred into protoplasts of LW-VF by PEG mediation [41], and the positive transformants were screened on a PDA containing 100 µg/mL hydomycin B and further verified by RT–PCR. The mixture was as follows: template cDNA 0.5 μL, specific primers 0.3 μL, 10×PCR mix 7.5 μL, ddH_2_O supplement to 15 μL. The amplification procedure: predenaturation at 95 °C for 3 min, denaturation for 30 s at 95 °C, annealing for 30 s at 55 °C, extension for 30 s at 72 °C, 35 cycles, elongation for 5 min at 72 °C. The temperature was lowered to 25 °C. The primer information is shown in Appendix A. 

### 2.7. Morphological Observation and Pathogenicity Analysis

Fresh mycelial agar plugs were punched from the colony margin of a 2-day-old culture of each strain, placed in the center of PDA medium, and cultured at 25 °C in the dark. There were at least three replicates for each strain. Colony morphology was observed every 24 h, and colony diameter was measured by crossing method until the whole dish was covered and photographed. Growth rate of each strain was calculated as follows: growth rate (cm/day) = (72 h diam. − 24 h diam.)/2. The pathogenicity analysis of strains was determined by inoculating branches of “Yuanhuang” and “Jinshui NO.1”. Branches with consistent growth were collected, washed and disinfected with 75% alcohol, wiped and dried with sterile water, which were divided for the same length. Inoculated the branches with fresh mycelial agar plugs and PDA medium as control, were placed on plates with plastic wrap to moisturize (90% relative humidity) at 25 °C, respectively. The lesions were measured and photographed after inoculation for 8 days.

### 2.8. Horizontal Transmission Assay

The contact culture trait of LW–C with BdCV1 and P3 transformant was assessed according to the method described earlier [40]. The LW–C served as the donor, and the P3 transformant as recipients, were co-cultured. The mycelium agar plugs from recipient P3 transformant strains for 7d were selected as candidate derived strains. The dsRNAs from the derived strains with abnormal colonies were extracted, and the band patterns of dsRNA was detected by electrophoresis, meanwhile the virulence status with BdCV1 was further verified by RT–PCR.

### 2.9. Stem-Loop RT-PCR

Referring to the method reported by Chen [42] as the basis for primer design, the mature sRNA sequence was cloned using stem-loop RT primer method commonly used in sRNA (small RNA molecules) studies. The primer information is shown in the Appendix A.

### 2.10. RT-qPCR

We used TRIzol (Aidlab, Beijing, China) method to extract and obtain good quality of total RNAs with a little modification based on manufacturer’s instructions. HiScrip Q RT SuperMix for qPCR (+gDNA Wiper) (Vazyme, Nanjing, China) was used for reverse transcription to obtain cDNA using 1.0 µg total RNA as template. ChamQ Universal SYBR qPCR Master Mix (Vazyme, Nanjing, China) was used for reverse transcription and template cDNA to perform qPCR, which is run using Bio-Rad IQTM5 Real-time System machine (Bio-Rad, Hercules, CA, USA). *N. benthamiana* GAPDH and *BdActin* were used as reference genes for corresponding quantitative analysis. Primer information is shown in Appendix A. Each sample was set three replicates. The obtained datas were analyzed by a comparative CT method (ΔΔCT) using the formula 2^−ΔΔCT^ [43].

### 2.11. cDNA Library Construction and RNA-Seq

LW-VF and P3 overexpression transformant’s hyphae were collected and cultured for 48 h, with two biological replicates, and sent to BGI corporation for extracting total RNA (BGI, Wuhan, China). Agilent 2100 Bioanalyzer was used to detect the RNA quality, and the results met the requirements of sequencing database construction. cDNA library construction and transcriptome sequencing were as follows: We used the method of magnetic beads with OligodT to enrich mRNAs containing poly(A) tail. The obtained RNA was segmented, then reverse-transcribed by random N_6_ primers. The ends of the synthetic dsDNA are patched flat, which is phosphorylated at the 5’ end. The linked products were amplified by PCR, then thermal-denatured to get single strand. A single strand circular DNA library is constructed using a designed bridge primer. The clean reads were mapped to *B. dothidea* LW-Hubei (GenBank: GCA_011064635.1) genome. Screening differentially expressed genes (DEGs) were accomplished by DNBSEQ with *p* ≤ 0.05, whose functional enrichment was performed by phyper function in R software (Version 4.2.2).

### 2.12. Data Analysis

The measured data were analyzed by *t* test in Graphpad (Version 8.0.2) software for significant difference (*p* < 0.05) and plotted.

## 3. Results

### 3.1. BdCV1-Encoded P3 Verification as a Potential Fungal RNA Silencing Suppressor in Exotic Plant System

To screen and estimate BdCV1-encoded proteins with function as putative fungal RNA silencing suppressor, the agro-infiltration system with GFP silencing suppression system was used for identification and analysis the spread of silencing signal [31,44].

#### 3.1.1. BdCV1-Encoded P3 Could Suppress the Local Silencing of GFP

Potential RNA silencing suppressor of BdCV1-encoded protein was screened and identified by tobacco transient transformation assay. P19, a strong RNA silencing suppressor, was used as positive control and pCNF3 as negative control. *Agrobacterium* expressing four ORFs of BdCV1 were mixed with 35S-GFP in equal volumes, respectively. At 3 days post inoculation (dpi) on leaves of *N. benthamiana* line 16c., the intensity of signal in each inoculation area was observed by using a portable long-wave ultraviolet lamp under dark conditions and photographed. The results showed that the signals in the area of leaves infiltrated with 35S-GFP and BdCV1-encoded RdRp, CP, P4 and pCNF3 as empty vector (EV) were weak, revealing that they could not inhibit GFP-induced RNA silencing. The BdCV1-encoded P3 infiltration with leaves exhibited strong and clear green fluorescence together comparable with that of positive control P19 (Appendix A).

To further confirm whether P3 had silencing inhibitory activity, *Agrobacterium* harboring 35S-GFP+P3, 35S-GFP+P19 and 35S-GFP+EV, respectively were infiltrated into different sites of the same leaf, and the fluorescence signals were observed at 3 days, which were consistent with the results of single leaf inoculation. Obvious fluorescence was observed in 35S-GFP+P19 infiltration zone, and green fluorescence was also observed in 35S-GFP+P3 infiltration zone. The fluorescence was weak in negative treatment (35S-GFP+EV), which further indicated that P3 had silence-inhibiting activity. Green fluorescence was still visible in 35S-GFP+P3 infiltrating area at 5 d (Figure 1A). The protein extracted from the infiltrated leaves was collected and GFP was detected by Western blot. The result showed that P3 and P19 infiltrated leaves had comparably strong protein band signals corresponding to the observed signal intensity, while the signals in EV leaves were weak at 5 dpi after inoculation (Figure 1B). Taken together, these data revealed that P3 could suppress GFP-induced local silencing.

#### 3.1.2. BdCV1-Encoded P3 Could Suppress dsRNA-Induced RNA Silencing in *N. benthamiana*

A mediator that induces RNA silencing to occur is dsRNA. Hence, we will investigate whether P3 can suppress dsRNA-mediated RNA silencing. In this study, 35S-GFP plus 35S-dsGFP was used as mediators to induce RNA silencing. GFP fluorescence signal could be observed at all the co-infiltrated area at 2 d. The signal of EV infiltrated area was weakened with the extension of time, while P19 and P3 were enhanced. At 4 d, obvious green fluorescence could be observed in the infiltrated area of P3 and P19, while there was weak fluorescence in EV infiltrated area (Figure 2A), which was consistent with those results at 5 d. The protein extracted from the infiltrated leaves was collected and GFP protein expression levels were detected and analyzed by Western blot (Figure 2B). It showed strong protein signals in P19 and P3, and weak protein signals in EV (Figure 2A,B). It indicated that P3 could inhibit RNA silencing signal induced by dsGFP.

#### 3.1.3. BdCV1-Encoded P3 Affected Cell to Cell Movement of Silencing Signals

In plants, once silencing is initiated in a single cell, the silencing signal can be transmitted between cells. We investigated whether P3 could inhibit the spread of intercellular RNA silencing by observing GFP signals at the edge of the infiltrating region. At 5 d, the expression of GFP in the area co-infiltrated with 35S-GFP and EV decreased significantly, and the red ring was observed around it. In contrast, no red ring was observed in both around the infiltrated area of positive control (35S-GFP + P19) and 35S-GFP + P3 (Figure 3). The GFP signal in adjacent cell regions of the infiltrating area was strongly reduced, and the infiltrating area showed an obvious red ring under UV light, just like that of the EV-infiltrated area. The results indicated that P3 can suppress the movement between cells of RNA silencing signal.

#### 3.1.4. BdCV1-Encoded P3 Could Suppress the Systemic Silencing of GFP in *N. benthamiana* Line 16c

In plants, silencing signals can move systematically. Hence, we examined whether P3 can suppress systemic silencing of GFP. The signal from the upper leaves was continuously observed, and the inoculation method was consistent with that before. The fluorescence of GFP in the upper leaves and stems of the plants was observed. With the extension of time, many plants containing 35S-GFP + EV showed red fluorescence. For 35S-GFP + P19, the majority of the plants exhibited green fluorescence. In contrast, all the plants infiltrated with 35S-GFP + P3 showed green fluorescence (Figure 4A). As shown in Table 1, the suppressing efficiency of 35S-GFP + P3 plants was nearly 100%, 35S-GFP + P19 plants was 48% and that of 35S-GFP + EV plants was about 20%, respectively (Table 1). The results suggest that P3 can powerfully suppress the systemic silencing of GFP in *N. benthamiana* line 16c.

Western blot analysis was used for detection GFP expression, after the upper leaves of *N. benthamiana* line 16c were infiltrated for 35 days. The results demonstrated that the signal of 35S-GFP + P19 and 35S-GFP + P3 treated leaves was strong, while no protein band visible to naked eyes was detected in EV-treated leaves by Western blot analysis (Figure 4B). It was further indicated that P3 could suppress the systematic silencing of GFP in plants.

### 3.2. P3 Induced the Leaf Necrosis Reaction

It was found that the leaves of 35S-GFP + P3 co-infiltrated *N. benthamiana* line 16c showed necrosis phenomenon. No such phenomenon was observed in the infiltrated area of 35S-GFP + P19 and 35S-GFP + EV (Figure 5A). The number of necrotic leaves was calculated and the necrosis rate was 100% (Appendix A). Further inoculation on wild type *N. benthamiana* showed that it could also produce necrosis, similar to that of *N. benthamiana* line 16c plants. Total RNA was extracted from inoculated leaves, and the expression levels of NbPR2, NbPR4 and NbPR5 were detected by RT–qPCR. According to data analysis, the expression levels of randomly selected three genes were significantly up-regulated with 2.0, 5.6 and 3.6 times, respectively after P3 infiltrating treatment in comparison to those of EV as the control group (Figure 5B). At inoculation 5 dpi, the infiltrated leaves were taken, and the color changes were observed by Trypan blue staining and DAB staining. After decolorization, it was found that Trypan blue staining exhibited blue color and DAB staining showed brown color in P3-infiltrated area, while no such color change was shown in those of the control group (EV) (Figure 5C,D). Taken together, our data revealed that P3 could induce the accumulation of reactive oxygen and stimulate cell necrosis reaction. It also showed that P3 induced up-regulation of PR gene expression and trigger plant defense response.

### 3.3. The Effects of P3 on Colony Growth and Pathogenicity of B. dothidea Strain

BdCV1 P3 was amplified from the cDNA of hypovirulent strain LW-C and cloned into vector pMD18-T (Appendix A). After transformation in *Escherichia coli* TOP 10, the P3 sequence was confirmed by sequencing. Then the P3 was ligated to the linearized vector pDL2 by homologous recombination to identify and obtain the recombinant plasmid designated as pDL2-P3 (Appendix A). In order to clarify the influence of P3 on host biological characteristics, P3 was transferred into *B. dothidea* LW-VF by protoplast transformation. Six positive transformants of P3 gene were obtained using PDA medium containing 100 μg/mL hygromycin and identified by RT–PCR (Appendix A), which indicated that P3 mRNA could effectively express at transcriptome level. The obtained positive transformants were named as OE1, OE2 and OE3, respectively used for further experimental analysis.

The transformants of P3 showed no significant differences from virus-free strain LW-VF in colony morphology and hyphal tip, while the colony of strain LW-C was abnormal (Figure 6A and Appendix A). Compared with LW-VF, the transformants of OE1, OE2 and OE3 strains had differences in the growth rate of mycelia and pathogenicity. The average growth rate of strain LW–VF was 2.42 cm/d, and the average growth rate of OE1, OE2 and OE3 were 1.80 to 2.13 cm/d with significantly lower than that of LW-VF (Figure 6A). The fresh mycelia of OE1, OE2 and OE3 and LW-VF strains were drilled and inoculated into the branches of one-year growing “Jinshui NO.1” pear. All strains could produce blank spots at 9 d. The average lesion length on pear branches inoculation with LW-VF, OE1, OE2 and OE3 were 2.97 cm and 1.30 to 1.85 cm, respectively, and showed significant differences (Figure 6B). Moreover, the pathogenicity results of inoculation on “Yuanhuang” branches were consistent with that of “Jinshui NO.1” (Appendix A). These results revealed that P3 could reduce the pathogenic virulence of *B. dothidea* strains.

### 3.4. The Effects of P3 on Gene Expression of B. dothidea Strain by Transcriptome Sequencing

To clarify the effects of P3 on *B. dothidea* host, we used RNA sequencing to analyze the differentially expressed genes (DEGs). Each stain obtained 4.36G of data. The clean reads from more than 96% raw reads were obtained after filtration, and the quality index met the requirements, by removing the raw reads (Appendix A). The standard for the screening of differentially expressed genes was *p* < 0.05, meanwhile significant differentially expressed genes selection criteria for *p* < 0.001 and |log2FC| ≥1 was set. The volcano map showed the distribution of differential genes in P3 overexpressed transformants and LW-VF (Figure 7A). Compared with LW-VF, there were 2289 up-regulated genes in P3 overexpression transformants, of which 733 were significantly different, and 271 were significantly different in 1787 down-regulated genes (Appendix A).

The partial gene information of P3 overexpression transformant and LW-VF with significant expressed differences was shown in Table 2. For example, gene GME9697_g belonged to the MFS transport family; GME4842_g was involved in ion transport. The genes GME1582_g and GME13620_g belonged to the glycoside hydrolase family and could participate in host pathogenicity. The genes GME13543_g, GME1096_g encoded the ABC transporter. The gene GME12401_g contains zinc finger domain, which was involved in regulating the growth, development and metabolism of fungi. GME1548_g encoded ribosomal protein L19, which was involved in DNA replication, repair and transcription. In conclusion, the overexpression of P3 regulated the expression of host genes related to transcription, transmembrane transport, pathogenicity and metabolism, respectively. The expression patterns of randomly selected 2 up-regulated and 5 down-regulated differentially expressed genes were analyzed by RT-qPCR, which were consistent with those of the transcriptome sequencing (Appendix A).

The result of GO functional annotation of differentially expressed genes showed that DEGs were annotated into three functional categories: biological process, cell component and molecular function. Biological processes were mainly involved in cellular processes, metabolic processes, localization and other aspects. Cell component were involved in cellular anatomical entity and protein-containing complex, while molecular functions of cell components were mainly in the aspects of binding, catalytic activity, transport activity and so on (Figure 7B).

### 3.5. P3 Is Involved in Host Gene Silencing Pathway

Since silencing suppressors can play a role by inhibiting the transcription of RNA silencing components, BdCV1 was transmitted into P3 transformants strains by contact culture to detect the expression level of RNA silencing components. OE1/LW–C, OE2/LW–C, OE3/LW–C and LW–C mycelia were collected and cultured for 5 d to extract total RNA. The results of RT-qPCR showed that the expression levels of *BdAGO1* and *BdDicer2* in OE1/LW–C, OE2/LW–C and OE3/LW–C were down-regulated compared with those of LW–C (Figure 8A,B), revealing that P3, as a silencing suppressor, may play a role by inhibiting the transcription of *BdAGO1* and *BdDicer2*.

The influence of P3 overexpression on vsiRNA was further analyzed. Four vsiRNAs with a large number of reads distributed on BdCV1 dsRNA3 were randomly selected for verification by stem–loop RT–PCR. Detailed information of vsiRNA is shown in Appendix A [45]. Agarose gel electrophoresis results showed that target fragments of about 100 bp were amplified in LW–C and P3/LW–C, but no corresponding fragments were amplified in LW–VF (Appendix A). The amplification products of 4 vsiRNA were obtained, cloned and sequenced, which were 100% consistent with the vsiRNAs nucleotide sequence corresponding to the small RNA sequencing analysis (Appendix A). The randomly selected four vsiRNAs express levels were determined by RT-qPCR. There were two up-regulated and two down-regulated expressions of vsiRNAs in OE1/LW–C, OE2/LW–C and OE3/LW–C (Figure 8C–F), indicating that P3 had an effect on the expression level of vsiRNA to a certain degree.

## 4. Discussion

Mycoviruses commonly infect hosts and play an important role in biocontrol due to their hypovirulent characteristics [46]. It was found that BdCV1 could cause *B. dothidea* pathogenicity attenuation, trigger host gene silencing, and have certain influences on host mRNA and sRNA expression, revealing that it participated in host interaction [16,17,47]. RNA silencing is an important host antiviral mechanism [48]. It is well known that plant viruses encode viral RNA silencing suppressor (VSR) to suppress host defense. However, the RNA silencing suppressor and its mechanism in fungal viruses has been less studied and explored the mechanism. So far, only mycoviruses CHV1-p29, CHV4-p24 and FgHV1-p20 in the *Hypoviridae*, MyRV3-s10 in the *Reoviridae*, and FgV1-ORF2 in the *Fusariviridae*, have been identified as silencing suppressors in fungi and exotic host *N. benthamiana* line 16c, respectively [33,34,35,36,37]. CHV1-p29 suppresses host RNA silencing based on inhibition the transcription of DCL2 and AGO2 [49]. p24 inhibits transcription of DCL2 and promotes stable infection of MyRV2 [34]. FgV1-ORF2 has the DNA-binding ability, which suppressed the transcriptional upregulation of FgDCL2 and FgAGO1, further to restrict host antiviral defense response, and promote viral multiplication in host [35]. In our study, BdCV1 P3 was identified to be an RNA silencing suppressor in tobacco system, which caused a decrease in host growth rate and reduced pathogenicity (Figure 1, Figure 2, Figure 4 and Figure 6).

Using the transient transformation system, P3 was firstly demonstrated to inhibit GFP-induced local silencing in *N. benthamiana* line 16c and the intercellular movement of RNA silencing (Figure 1 and Figure 3). It was further confirmed that P3 could inhibit the upward transmission of RNA silencing signal, thus inhibiting systemic RNA silencing (Figure 4). Moreover, we found that P3 could inhibit the RNA silencing induced by dsRNA (Figure 2). Therefore, we hypothesized that P3 play a role in the downstream stage of dsRNA formation, and its specific regulatory function as an RNA silencing inhibitor in fungi and its participation in virus-host interaction need to be further explored. P3 can induce the necrosis of tobacco cells, the accumulation of reactive oxygen species and induce up-regulation of PR gene expression related to plant defense response (Figure 5). Over-expressed P3 can further reduce host pathogenicity, our hypothesis is that it can induce plant immunity to resist pathogen infection. In addition, previous RNA sequencing and bioinformatics results revealed that BdCV1 activated and regulated the expression level of resistance genes in pear with LW–C and LW–1 in comparison with those of LW–P and LW–VF (Unpublished results). It was reported that mycovirus-induced hypervirulence of *Leptosphaeria biglobosa* enhances systemic acquired resistance to *Leptosphaeria maculans* in *Brassica napus* [50]. It is supposed that whether BdCV1-P3 can be recognized as an effector alone and cause the necrosis of plant cells to trigger plant defense response to inhibition pathogen infection, need be further verified.

RNA silencing components had different effects on host growth, development and pathogenicity. For example, when *ago1* and *dcl1* were knocked out in *Colletotrichum higginsianum*, the sporulation rate of the strain decreased, conidial morphology changed, and the numbers of small RNAs increased [51]. AGO2 and DCL2 are involved in the growth of mycelium in *Valsa mali*. AGO3 is related to the formation of sporulation, while DCL1, AGO2 and AGO3 are related to pathogenicity [52,53,54]. It was also confirmed that BdAGO1, BdAGO3 and BdDicer2 had regulatory effects on the growth and pathogenicity of *B. dothidea* [18,45]. It reported that vsiRNA inhibited host RNA silencing by acting as a decoy sRNA to promote viral infection. Studies have verified the function of vsiRNA in targeting viral RNA or DNA and host transcription. Silencing suppressors evolved by plant viruses mainly counteract the antiviral silencing process by preventing functional vsiRNA-RISC assembly [55,56]. The results showed that P3 expression down-regulated the expressions of BdAGO1 and BdDicer2 (Figure 8A,B), which were related to gene silencing, and also affected the expression of vsiRNA (Figure 8C). It was speculated that P3 was involved in the host gene silencing pathway, providing molecular information and a theoretical basis for P3 as potential silencing suppressor in fungi.

Next, we elucidated the effect of P3 on the fungi host. The colony morphology, growth rate, and pathogenicity of each transformant were measured. It was clear that the P3 protein could slow down the growth rate and weaken the virulence (Figure 6). The effects of BdCV1 P3 on host were clarified, which provided molecular information for further analysis of the function of BdCV1. Transcriptome sequencing analysis showed that P3 affected host genes expression involved in pathogenesis, transcription, transmembrane and metabolic pathways. For example, the protein encoded by gene GME1582_g is glycoside hydrolase family 62, which was involved in carbohydrate metabolism. Gene GME12401_g contains a zinc finger domain, which can be involved in regulating the growth, development and metabolism of fungi. When BR1 transcription factor was knocked out in *Fusarium graminearum*, the growth rate and virulence of the strain were decreased, and the number of conidia was significantly reduced [57,58]. The gene GME1096_g encoded the ABC transporter, a virulence factor in *Botrytis cinerea* that increased pathogen resistance to plant antitoxins and fully restored virulence in hosts lacking this plant antitoxin [59]. P3 expression down-regulates the expression of these genes, and it is speculated that P3 was involved in gene interaction to reduce host virulence.

In conclusion, this is the first identification of BdCV1 P3 as a silencing suppressor in the agro-infiltration assay. The expression of P3 has a certain effect on RNA silencing components and vsiRNAs, which provides key reference information for exploration of P3 as the roles of silencing suppressor in fungi. Furthermore, overexpression of P3 could inhibit the growth and decrease the pathogenicity of *B. dothidea*. Combined with transcriptome sequencing results, P3 affected the expression of host genes involved in pathogenesis, transcription, transmembrane transport and metabolic processes. These results laid the foundation for clarifying the potential application of mycovirus in biological control of pear ring rot disease.

## Figures and Tables

**Figure 1 cells-12-02386-f001:**
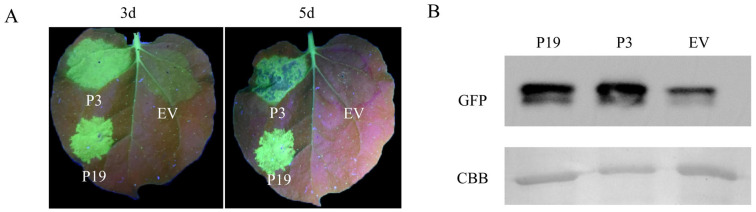
BdCV1-encoded P3 inhibited local silencing of GFP. (**A**) *N. benthamiana* line 16c leaves infiltrated with a mixture of *Agrobacterium* cultures containing 35S-GFP + P19 or the EV were positive or negative control, respectively. GFP signal was observed at 3 d–5 d and taken photographs under UV light. (**B**) Western blot was used for analysis of GFP protein expression in infiltrated leaves at 5 d.

**Figure 2 cells-12-02386-f002:**
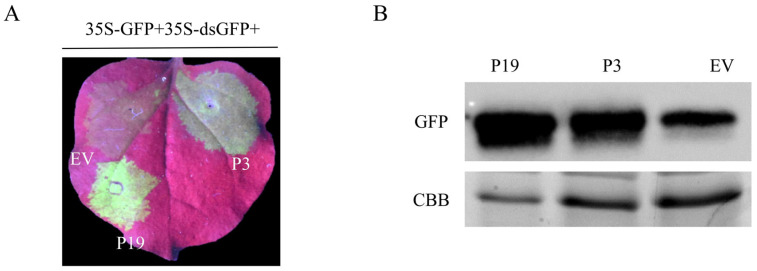
P3 can suppress dsGFP-mediated silencing in *N. benthamiana*. (**A**) *N. benthamiana* leaves infiltrated with a mixture of *Agrobacterium* cultures containing 35S-GFP plus P19 or EV were positive or negative control, respectively. GFP signal was observed at 4 d and taken photographs under UV light. (**B**) Western blot was used for analysis of GFP protein expression in infiltrated leaves at 5 d.

**Figure 3 cells-12-02386-f003:**
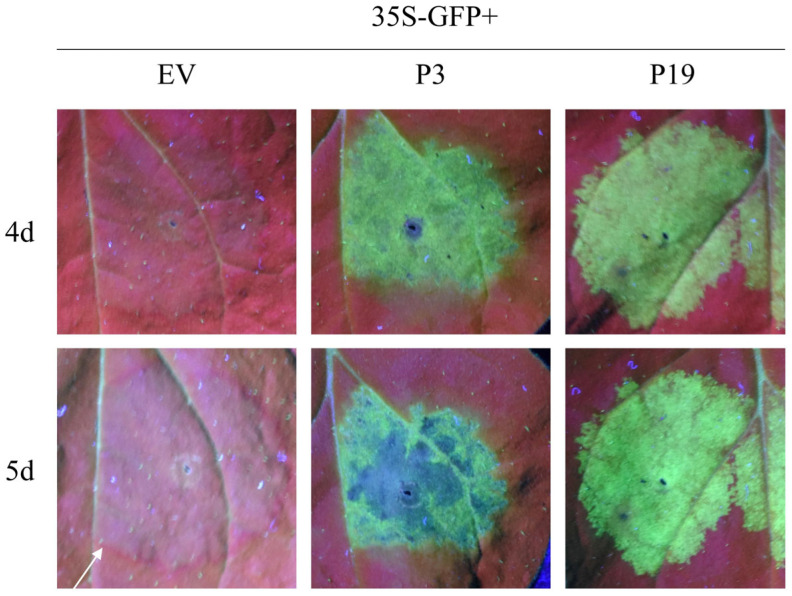
P3 could inhibit short-distance spread of the GFP silencing signal in *N. benthamiana* line 16c. Leaves infiltrated with a mixture of *Agrobacterium* cultures containing 35S-GFP + P19 or the EV were positive or negative control, respectively. Pictures were taken under UV light at 4 d and 5 d. The white arrow points to red ring, the characteristics of movement between cells of silencing signals.

**Figure 4 cells-12-02386-f004:**
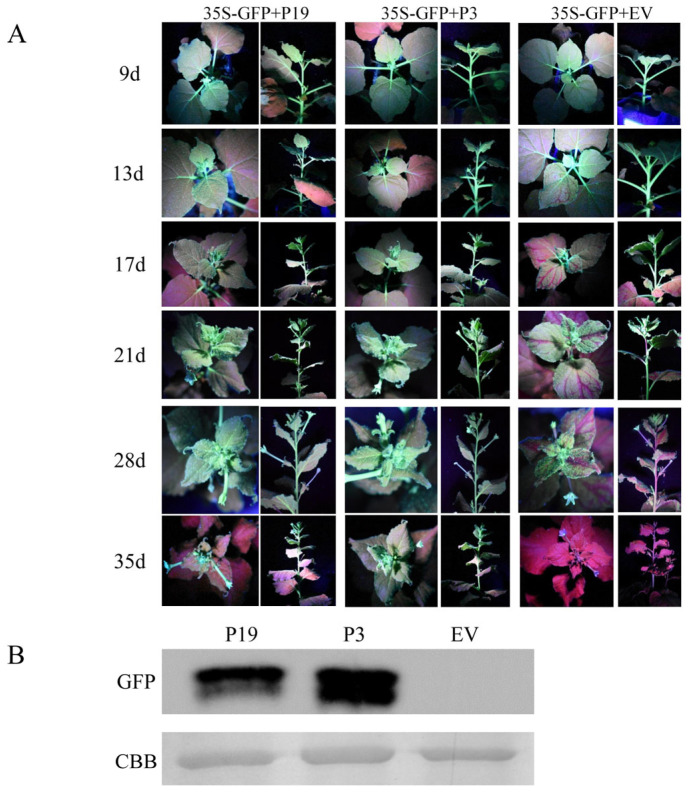
Systemic silencing suppression mediated by BdCV1-encoded P3. (**A**) *N. benthamiana* line 16c upper leaves infiltrated with a mixture of *Agrobacterium* cultures containing 35S-GFP + P19 or EV were positive or negative control, respectively. GFP signal was observed and taken photographs under UV light. (**B**) Western blot analysis was used for detection the expression level of GFP protein in infiltrated leaves at 35 d.

**Figure 5 cells-12-02386-f005:**
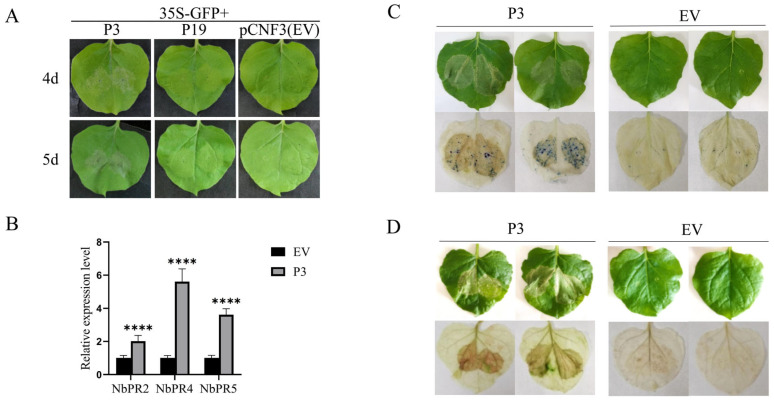
P3 induced the leaf necrosis reaction. (**A**) P3 induced the leaf necrosis reaction of *N. benthamiana* line 16c leaves. (**B**) Effect of P3 expressed on the transcript levels of NbPR2, NbPR4 and NbPR5. *N. benthamiana* GAPDH served as reference gene. Results are mean ± SD calculated. Significant difference analysis was performed by *t*-test, **** *p* < 0.0001. (**C**) The phenotype of *N. benthamiana* after Trypan blue staining. (**D**) The phenotype of *N. benthamiana* after DAB staining.

**Figure 6 cells-12-02386-f006:**
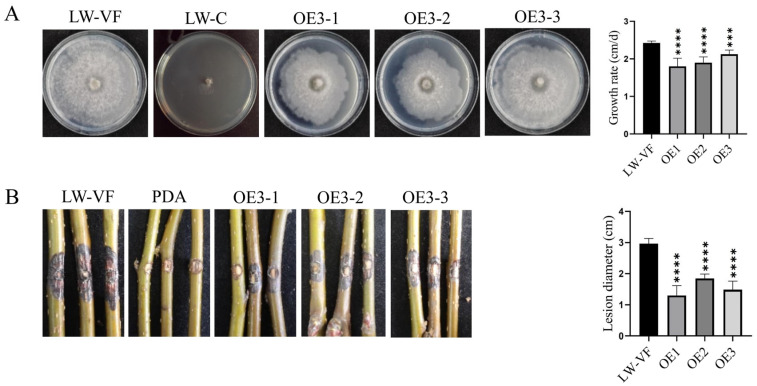
The effects of P3 on colony growth and pathogenicity of *B. dothidea* strains. (**A**) The colonial morphology and growth rate of strains LW–VF, LW–C and P3 overexpresses transformants at 3 days. (**B**) The pathogenicity of each strain on “Jinshui NO.1” branches inoculated for 9 days. Results are mean ± SD calculated. Significant difference analysis was performed by *t*-test, *** *p* < 0.001, **** *p* < 0.0001.

**Figure 7 cells-12-02386-f007:**
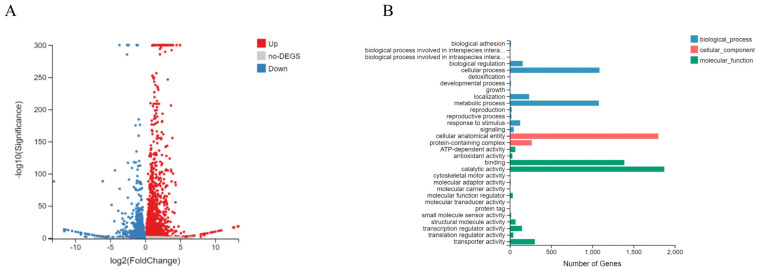
Volcano plot (**A**) and GO functional annotation (**B**) of DGEs between the P3 overexpressing transformants and LW–VF.

**Figure 8 cells-12-02386-f008:**
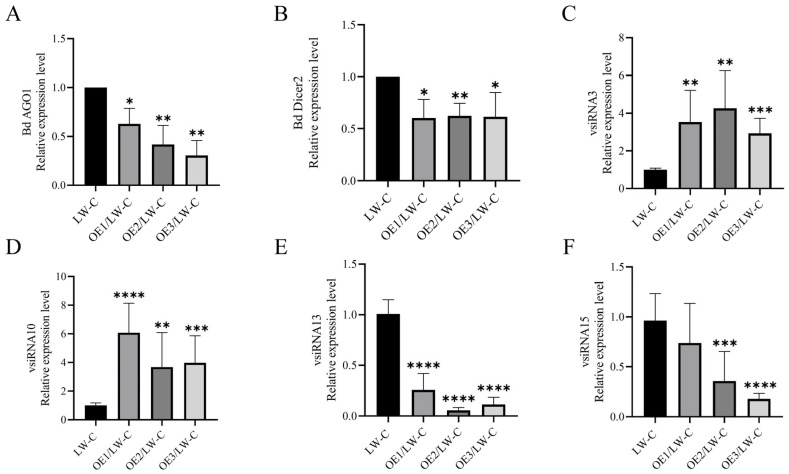
Effect of P3 on RNA silencing components and vsiRNA expression levels. The expression level of Bd AGO1 (**A**), Bd Dicer2 (**B**) and vsiRNAs (**C**–**F**) in OE1/LW–C, OE2/LW–C OE3/LW–C, and LW–C strains, respectively, were detected by RT–PCR. *BdActin* served as reference gene. Results are mean ± SD calculated. Significant difference analysis was performed by *t*-test, * *p* < 0.05, ** *p* < 0.01, *** *p* < 0.001, **** *p* < 0.0001.

**Table 1 cells-12-02386-t001:** The statistics of systemic silencing suppression efficiency.

Infiltration Treatment	Number of Treated Plants	Number of Systemic Silencing Plants	Suppressing Efficiency (%)
35S-GFP + P19	25	13	48%
35S-GFP + P3	25	0	100%
35S-GFP + EV	25	20	20%

**Table 2 cells-12-02386-t002:** Information of partial DEGs between the P3 overexpressing transformants and LW–VF.

Gene ID	log2FC	Regulation	Function
GME9697_g	10.69	Up	Major facilitator superfamily
GME4842_g	2.242	Up	Calcium uniporter protein, mitochondrial
GME7336_g	2.203	Up	Zinc finger C2H2-type/integrase DNA-binding protein
GME9654_g	2.754	Up	Zinc finger C_2_H_2_-type protein
GME10313_g	1.499	Up	Phospholipid/glycerol acyltransferase
GME10276_g	2.094	Up	Cytochrome P450
GME3341_g	9.696	Up	Putative serine threonine-protein phosphatase pp-z protein
GME2378_g	9.497	Up	Putative nitrosoguanidine resistance protein
GME4519_g	8.885	Up	Alpha/beta hydrolase fold-1
GME9641_g	7.047	Up	Short-chain dehydrogenase/reductase SD
GME7258_g	6.767	Up	Serine hydrolas
GME9987_g	−3.869	Down	RNA-directed DNA polymerase (reverse transcriptase)
GME1582_g	−5.472	Down	Putative glycoside hydrolase family 62 protein
GME13543_g	−3.241	Down	ABC1 family protein
GME3679_g	−7.541	Down	Putative allantoinase protein
GME8425_g	−11.56	Down	Alcohol dehydrogenase superfamily zinc-containing
GME596_g	−11.33	Down	Oxoglutarate/iron-dependent oxygenase
GME1096_g	−9.197	Down	ABC1 family protein
GME12824_g	−8.550	Down	Nucleic acid binding OB-fold tRNA/helicase-type
GME1548_g	−8.240	Down	Ribosomal protein L19
GME9360_g	−7.578	Down	Fungus-specific transcription factor domain-containing protein
GME11114_g	−6.576	Down	FAD-binding domain-containing protein
GME12401_g	−6.076	Down	Putative C6 zinc finger domain-containing protein
GME13620_g	−4.800	Down	Glycosyltransferase family 1 protein

## Data Availability

All data is available in Appendix A.

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
