# Peer review of "BdCV1-Encoded P3 Silencing Suppressor Identification and Its Roles in Botryosphaeria dothidea, Causing Pear Ring Rot Disease"

_cells, 2023, doi:10.3390/cells12192386_

Round 1

Reviewer 1 Report

Dear Authors,

I want to congratulate you. In my opinion, the manuscript is of significant scientific interest, due to the possible biotechnological application to reduce pathogens' aggressiveness. All the experimental procedures are sound and the results properly support the final outcome.

Only the quality of English needs to be improved, as specified in the related Comments. I strongly suggest a language revision by a native speaker.

Dear Authors,

as already mentioned, many parts of the text need to be improved, especially in the use of proper verbal tenses, in the Introduction, M&M, RNASeq paragraph 3.4, and Discussion. In my opinion, a thorough revision of the entire paper by a native speaker is strongly recommended

Plus, all scientific names must be in italics, even for the viruses, including those in the References.

Below are some (not exhaustive) detailed comments:

Below are some (not exhaustive) detailed comments:

Line 17: No need for the question mark

Lines 19-26: Please, revise the sentence to make it clearer. 

Line 30 (and the rest of the text): No need for past tense, please use the present tense.

Line 33: “a kind of” sounds quite weird. Please, revise the sentence.

Line 36: Add space in "insects[5]".

Line 37 (and the rest of the text): Agaricus bisporus must be in italics.

Line 39: Aspergillus fumigatuS: the last "s" is missing.

Line 57: “The main influence of host fungi on viruses in RNA silencing” The sentence is not clear. Please, revise it.

Line 85: Add a reference for treatments and chemical control.

The "Materials and methods" section requires an extensive revision in both English style and grammar.

Line 98: remove “which”

Lines 121-122: pDL2 vector donated from Prof… move this sentence to the Acknowledgement.

Line 125: Vector construction "IS" shown in Table S1, not in past tense.

Line 127: “Sucking” is not the proper word, please revise it.

Line 137: "the vectors were obtained".

Paragraph 2.4: ProteinS extraction, since you mention “total”. Adjust the verbal tense, the proteins were extracted for Western blot...

Line 150: in per well?

Line 181: please, add the PCR reaction and program.

Line 208: sRNA? Explicit the first time you mention it.

Line 244: Which tool was used for clustering? DNBSEQ too?

Line 246: add the version of Graphpad used

Line 283: check the scientific name in the caption and make it Italic

Line 290: “Further” is not the proper word to start the sentence. “Hence” could be better.

Line 332: same as line 290

Lines 375-377: please revise. The results showed that P3 could trigger the accumulation of … It also showed that P3 induced up-regulation…

Line 387: veCtor

As said, paragraph 3.4 is not clear, especially for those who are not familiar with this kind of RNAseq analysis. Please revise.

Line 468: please introduce vsiRNA acronym the first time you mention it.

Revise the Discussion and carefully check the verbs and the verbal tense.

Line 526: Valsa mali

Line 538: "further" is not the proper word.

Line 666: can you cite a Master thesis? Please, check it with the Journal Editor

Author Response

A point-by-point response to the comments from the reviewer

Reviewer #1 comments and suggestions:

I want to congratulate you. In my opinion, the manuscript is of significant scientific interest, due to the possible biotechnological application to reduce pathogens' aggressiveness. All the experimental procedures are sound and the results properly support the final outcome. Only the quality of English needs to be improved, as specified in the related Comments. I strongly suggest a language revision by a native speaker.

Answer: We greatly appreciate the positive and constructive comments, and the English editing from this reviewer that have improved the manuscript. We noticed that some of the descriptions in the previous version were not accurate and/or complete. We have corrected this in the revised manuscript according to your suggestions. The revised manuscript has been thoroughly checked by a native English-speaking colleague.

Comments on the Quality of English Language

Dear Authors,

as already mentioned, many parts of the text need to be improved, especially in the use of proper verbal tenses, in the Introduction, M&M, RNASeq paragraph 3.4, and Discussion. In my opinion, a thorough revision of the entire paper by a native speaker is strongly recommended Plus, all scientific names must be in italics, even for the viruses, including those in the References.

Answer: We have now worked on both language and readability, and have also involved native English speakers for language corrections. We really hope that the flow and language level have been substantially improved. And part scientific names are in italics, including the references.

Minor points are shown below.

  1. Line 17: No need for the question mark

Answer: Thank you your insight suggestion. We have rephrased this sentence.

  1. Lines 19-26: Please, revise the sentence to make it clearer.

Answer: Thank you for reminding us of the inappropriate writing. It was described in detail in the revised manuscript.

  1. Line 30 (and the rest of the text): No need for past tense, please use the present tense.
  2. Line 33: “a kind of” sounds quite weird. Please, revise the sentence.
  3. Line 36: Add space in "insects[5]".
  4. Line 37 (and the rest of the text): Agaricus bisporus must be in italics.
  5. Line 39: Aspergillus fumigatuS: the last "s" is missing.
  6. Line 85: Add a reference for treatments and chemical control.
  7. Line 98: remove “which”
  8. Line 125: Vector construction "IS" shown in Table S1, not in past tense.
  9. Line 127: “Sucking” is not the proper word, please revise it.
  10. Line 137: "the vectors were obtained".
  11. Line 244: Which tool was used for clustering? DNBSEQ too?
  12. Line 387: veCtor
  13. Line 526: Valsa mali
  14. Line 538: "further" is not the proper word.

Answer: Thank you for reminding us of the inappropriate writing. They were changed into appropriate expressions at the corresponding position in the revised manuscript as you suggested.

  1. Line 57: “The main influence of host fungi on viruses in RNA silencing” The sentence is not clear. Please, revise it.

Answer: Thank you for your insight suggestion. They were changed into appropriate expressions at the corresponding position in the revised manuscript.

  1. Lines 121-122: pDL2 vector donated from Prof… move this sentence to the Acknowledgement.

Answer: Thank you for your constructive suggestion. We have moved this to the acknowledgement.

  1. Paragraph 2.4: ProteinS extraction, since you mention “total”. Adjust the verbal tense, the proteins were extracted for Western blot...

Answer: Thank you for reminding us of the inappropriate writing. It was described in detail in the revised manuscript.

  1. Line 150: in per well?

Answer: We apologize for the incorrect statement, which has now been corrected to" in every gel lane”.

  1. Line 181: please, add the PCR reaction and program.

Answer: Thank you for your constructive suggestion. It was described in detail in the revised manuscript.

  1. Line 208: sRNA? Explicit the first time you mention it.
  2. Line 468: please introduce vsiRNA acronym the first time you mention it.

Answer: Thank you for reminding us of the inappropriate writing. We have introduced sRNA and vsiRNAs at the first time we mention them.

  1. Lines 375-377: please revise. The results showed that P3 could trigger the accumulation of … It also showed that P3 induced up-regulation…

Answer: Thank you for reminding us of the inappropriate writing. We have modified it as your suggestion.

  1. The "Materials and methods" section requires an extensive revision in both English style and grammar.
  2. Revise the Discussion and carefully check the verbs and the verbal tense.

Answer: Thank you for your constructive suggestion. We have fixed language-related problems. We hope this part of the language to be improved.

  1. As said, paragraph 3.4 is not clear, especially for those who are not familiar with this kind of RNA-seq analysis. Please revise.

Answer: Thank you for your insight suggestion. We have revised this part so hopefully it's easy to understand.

 28. Line 666: can you cite a Master thesis? Please, check it with the Journal Editor

Answer: I checked the reference. The Master thesis can be cited in the manuscript.

Reviewer 2 Report

The manuscript is well-written and organized. It presents the results of original research and makes a valuable contribution to the knowledge of silencing inhibitory activity of BdCV1 encoded proteins, and authors identified BdCV1 P3 as a BdCV1 RNA silencing suppressor in Nicotiana benthamiana line 16c.

All sections of the manuscript are properly organized. The methodology is well elaborated and presented in detail. Results are presented in a clear way throughout the text, as well as in figures and tables. The discussion is presented in a proper way and supported with the adequate citations from the literature.

Specific remarks are given in the attached document.

Some comments are given in the attached document.

Author Response

A point-by-point response to the comments from the reviewer

Reviewer #2 comments and suggestions:

The manuscript is well-written and organized. It presents the results of original research and makes a valuable contribution to the knowledge of silencing inhibitory activity of BdCV1 encoded proteins, and authors identified BdCV1 P3 as a BdCV1 RNA silencing suppressor in Nicotiana benthamiana line 16c.

All sections of the manuscript are properly organized. The methodology is well elaborated and presented in detail. Results are presented in a clear way throughout the text, as well as in figures and tables. The discussion is presented in a proper way and supported with the adequate citations from the literature. Specific remarks are given in the attached document.

Answer: We greatly appreciate the positive and constructive comments on the manuscript from you.

Minor points from the attached document are listed below.

  1. Line 16-17: The aim of our study was to investigate...
  2. Add space in "insects[5]".
  3. Line 48: First mentioning in the text ( dothidea).
  4. Line 84: First mentioning in the text ( benthamiana)
  5. Line 150: Do not start a sentence with number.
  6. Line 152: Needlessly.
  7. Line 165: Delete “to be dyed” and revise it.
  8. Line 173: Delete “The results were observed and recorded by photograph”.
  9. Line 180: No need for past tense, please use the present tense.
  10. Line 165: Delete “ cross ”.
  11. Line 199: “previous method” revised to “the method described earlier”.
  12. Line 209-212: General description. delete, unnecessary.
  13. Line 224-225: “amplification system” revised to “mixture”.
  14. Line 237: 6 should subscript.
  15. Line 508: “it is speculated” revised to “we hypothesized”.
  16. Line 516: Add source. Unpublished results or published results? (data not show)
  17. Line 518: Not italic, add space before.
  18. Line 526: Valsa mali

Answer: Thank you for reminding us of the inappropriate writing. They were changed into appropriate expressions at the corresponding position in the revised manuscript as you suggested.

  1. Line 359-361: This part should be moved to discussion section.

Answer: Thank you for your constructive suggestion. According to your suggestion, we have moved it to the introduction part.

  1. Line 144: Correct English in this sub-chapter 2.4.

Answer: We have now revised it in the corresponding parts of this sub-chapter 2.4.

  1. Line555-556: Correct grammar. Example: This is the first identification of...

Answer: We have now revised it in the corresponding parts.

  1. Line 513: Some reference for this speculation? is this only your statement? if so: our hypothesis is that it can induce...

Answer: Thank you for your constructive suggestion. This speculation is our statement based on the obtained results. We have standardized the description as you suggestion.

Reviewer 3 Report

Dear Authors,

Review of the paper entitled: "BdCV1-Encoded P3 Silencing Suppressor Identification and Its Roles in Botryosphaeria dothidea, the Causal Agent of Pear Ring Rot Disease."

1.     The title of the work is clear and precise. It clearly indicates the topic of the study, which is the identification of the P3 suppressor encoded by BdCV1 and its role in the case of Botryosphaeria dothidea, the pathogen causing pear ring rot disease. Overall, the title is solid and does its job of pointing out the topic of the study and informing readers about its essence. However, if possible, consider shortening the title to make it even more concise but still informative.

2.     The introduction is very clear and precise, providing the reader with the necessary context regarding the research problem. Several key elements are well covered:

-      a)Disease context:

-      b) Need for control:

-      c) The role of fungal viruses

-      d) Previous studies:

-      e) Purpose of the study: The final part of the introduction clearly states the goal of the study, which is to identify and understand the role of the BdCV1-encoded P3 suppressor and its impact on host defence and gene suppression pathways.

The introduction is well-written and readable, introducing the reader to the topic of the study and motivating further reading. However, it could be made even more concise by eliminating some details about other fungal viruses to focus more on the study of BdCV1. It provides a solid basis for further understanding of the issues and research goals of the work.

3.     Material and methods. The introduction is very clear and precise, providing the reader with the necessary context regarding the research problem. Several key elements are well covered:

-      Context of the disease: The introduction begins by introducing the reader to the problem of pear ring rot disease, caused by Botryosphaeria dothidea. The effects of this disease are explained, including its impact on pear trees and the pear industry.

-      Need for control: The text emphasizes the importance of finding "green" methods to control this disease, suggesting that existing methods may have a negative impact on the environment.

-      Role of fungal viruses: The introduction discusses the importance of fungal viruses, both in terms of pathogenicity and potential use as a biological control tool.

-      Previous research: The authors present their previous research, including the identification of BdCV1 as a weak pathogenicity agent and the description of the BdCV1 genome.

-      Purpose of the study: The final part of the introduction clearly states the goal of the study, which is to identify and understand the role of the P3 suppressor encoded by BdCV1 and its impact on host defence and gene suppression pathways.

The introduction is well-written and clear, introducing the reader to the topic of the study and motivating further reading. However, it could be made even more concise by eliminating some details about other fungal viruses to focus more on the study of BdCV1. Overall, however, it provides a solid basis for further understanding of the issues and research goals of the work.

4.     Material and Methods – This chapter is extensive and contains many details about laboratory and experimental procedures. The Description of Materials and Methods is very detailed and contains relevant information. However, some passages could be shortened slightly to maintain a more concise text structure.

5.     RESULTS - this section discusses the discovery of BdCV1-encoded P3 as a potential suppressor of fungal RNA silencing in an exotic plant system. This study used an agroinfiltration system with a GFP silencing suppression system to identify and analyse the spread of silencing signals.

6.     The Discussion chapter discusses the results of experiments on the role of the P3 protein of the BdCV1 virus in interaction with the plant and the impact on the development and pathogenicity of the fungus B. dothidea.

7.     Key conclusions concerned aspects of work such as:

-      Local mute suppression:

-      Suppression of dsRNA-induced silencing:

-      Inhibition of cell-to-cell spread:

-      Systemic mute suppression:

-      Induction of leaf necrosis:

-      Effect on colony growth and pathogenicity:

-      Transcriptome analysis

-      Inhibition of RNA silencing components:

-      Effect on vsiRNA.

8.     The conclusions of this chapter suggest that the P3 protein of BdCV1 plays an important role in virus-host interaction. It inhibits RNA-silencing, affects the pathogenicity of the fungus B. dothidea and regulates the expression of host genes. This discovery may be important both in understanding the mechanisms of viral infections and in the development of biological pest control strategies. However, further research is needed to better understand the details of P3 function and its effects on plants and fungi.

Minor English editing is required.

Author Response

A point-by-point response to the comments from the reviewer

Reviewer #3 comments and suggestions:

Dear Authors,

Review of the paper entitled: "BdCV1-Encoded P3 Silencing Suppressor Identification and Its Roles in Botryosphaeria dothidea, the Causal Agent of Pear Ring Rot Disease."

  1. The title of the work is clear and precise. It clearly indicates the topic of the study, which is the identification of the P3 suppressor encoded by BdCV1 and its role in the case of Botryosphaeria dothidea, the pathogen causing pear ring rot disease. Overall, the title is solid and does its job of pointing out the topic of the study and informing readers about its essence. However, if possible, consider shortening the title to make it even more concise but still informative.
  2. The introduction is very clear and precise, providing the reader with the necessary context regarding the research problem. Several key elements are well covered:
  3. a) Disease context:
  4. b) Need for control:
  5. c) The role of fungal viruses
  6. d) Previous studies:
  7. e) Purpose of the study: The final part of the introduction clearly states the goal of the study, which is to identify and understand the role of the BdCV1-encoded P3 suppressor and its impact on host defence and gene suppression pathways.

The introduction is well-written and readable, introducing the reader to the topic of the study and motivating further reading. However, it could be made even more concise by eliminating some details about other fungal viruses to focus more on the study of BdCV1. It provides a solid basis for further understanding of the issues and research goals of the work.

  1. Material and methods. The introduction is very clear and precise, providing the reader with the necessary context regarding the research problem. Several key elements are well covered:

Context of the disease: The introduction begins by introducing the reader to the problem of pear ring rot disease, caused by Botryosphaeria dothidea. The effects of this disease are explained, including its impact on pear trees and the pear industry.

Need for control: The text emphasizes the importance of finding "green" methods to control this disease, suggesting that existing methods may have a negative impact on the environment.

Role of fungal viruses: The introduction discusses the importance of fungal viruses, both in terms of pathogenicity and potential use as a biological control tool.

Previous research: The authors present their previous research, including the identification of BdCV1 as a weak pathogenicity agent and the description of the BdCV1 genome.

Purpose of the study: The final part of the introduction clearly states the goal of the study, which is to identify and understand the role of the P3 suppressor encoded by BdCV1 and its impact on host defence and gene suppression pathways.

The introduction is well-written and clear, introducing the reader to the topic of the study and motivating further reading. However, it could be made even more concise by eliminating some details about other fungal viruses to focus more on the study of BdCV1. Overall, however, it provides a solid basis for further understanding of the issues and research goals of the work.

  1. Material and Methods – This chapter is extensive and contains many details about laboratory and experimental procedures. The Description of Materials and Methods is very detailed and contains relevant information. However, some passages could be shortened slightly to maintain a more concise text structure.
  2. RESULTS - this section discusses the discovery of BdCV1-encoded P3 as a potential suppressor of fungal RNA silencing in an exotic plant system. This study used an agroinfiltration system with a GFP silencing suppression system to identify and analyse the spread of silencing signals.
  3. The Discussion chapter discusses the results of experiments on the role of the P3 protein of the BdCV1 virus in interaction with the plant and the impact on the development and pathogenicity of the fungus B. dothidea.
  4. Key conclusions concerned aspects of work such as:

Local mute suppression:

Suppression of dsRNA-induced silencing:

Inhibition of cell-to-cell spread:

Systemic mute suppression:

Induction of leaf necrosis:

Effect on colony growth and pathogenicity:

Transcriptome analysis

Inhibition of RNA silencing components:

Effect on vsiRNA.

  1. The conclusions of this chapter suggest that the P3 protein of BdCV1 plays an important role in virus-host interaction. It inhibits RNA-silencing, affects the pathogenicity of the fungus dothidea and regulates the expression of host genes. This discovery may be important both in understanding the mechanisms of viral infections and in the development of biological pest control strategies. However, further research is needed to better understand the details of P3 function and its effects on plants and fungi.

Answer: We greatly appreciate the positive and constructive comments from you. According to your suggestions, we have shortened the title to make it more concise but still informative. Meanwhile, the description of other fungal viruses in the introduction has been modified by deleting some details to make more concise and focus more on the study of BdCV1 as suggested. In addition, we appropriately reduced the lengthy description of materials and methods to make it concise.

In addition, you referred“The text emphasizes the importance of finding "green" methods to control this disease, suggesting that existing methods may have a negative impact on the environment”, we added the information (“At present, the main control methods for pear ring rot are bagging treatment and chemical control with a negative impact on the environment”) in the revised manuscript.
